# Nursing Staff's Observations of BPSD Amongst Older Adults with Dementia Living in a Nursing Home: A Qualitative Study

Emerentia Grootscholten [1,*], Irina Poslawsky [2] and Ton Bakker [1]

1 Research Centre Innovations in Care, Rotterdam University of Applied Sciences, Rochussenstraat 198, 3015 EK Rotterdam, The Netherlands
2 Masteropleiding Klinische Gezondheidswetenschappen, Utrecht University, Heidelberglaan 100, 3584 CX Utrecht, The Netherlands
* Correspondence: e.c.m.grootscholten@hr.nl; Tel.: +31-6-8018-1200

**Abstract:** The majority of older adults with dementia living in a nursing home exhibit behavioral and psychological symptoms of dementia (BPSD). This behavior is difficult for residents to cope with. Early recognition of BPSD is important in order to implement personalized integrated treatment, and nursing staff are in the unique position to consistently observe residents' behavior. The aim of this study was to explore nursing staff's experiences observing BPSD of nursing home residents with dementia. A generic qualitative design was chosen. Twelve semi-structured interviews were conducted with nursing staff members until data saturation. Data were analyzed using inductive thematic analysis. Four themes were identified: "group harmony" observations from a group perspective, focused on the disturbance of group harmony; an "intuitive approach", which involves observing unconsciously and without a set method; "reactive intervention", which refers to immediate removal of observed triggers without exploring the causes of behaviors; and "sharing information", which is delayed sharing of observed behavior with other disciplines. The current way in which nursing staff observe BPSD and share observations within the multidisciplinary team explain several existing barriers to achieving high treatment fidelity for BPSD with personalized integrated treatment. Therefore, nursing staff must be educated to structure their daily observations methodologically and interprofessional collaboration improved to share their information in a timely manner.

**Keywords:** observation; BPSD; dementia; nursing staff; nursing home; qualitative study



## 1. Introduction

Dementia is a disease that primarily affects older adults [1]. People with dementia increasingly depend on others for assistance with daily living due to progressive cognitive, functional, and behavioral deterioration [2]. Therefore, in the Netherlands, approximately 35% of all older adults with dementia reside in nursing homes [1] defined as a domestic-style environment providing 24-hour care for residents [3]. Studies have shown that 75–91% of nursing home residents with dementia have clinically significant behavioral and psychological symptoms of dementia (BPSD). The prevalence of the most predominant BPSD are as follows: depression (43.9%), apathy (43.1%), and agitation/aggression (31.2%) [4]. Behavioral and psychological disturbance impair the quality of life of older adults with dementia [5] and distress nursing staff [6]. Various terms are used in the literature to describe behavior caused by BPSD, such as "responsive behavior", "disruptive behavior", "behaviors of concern", "problem behavior", and "challenging behavior" [7]. Person-centered care (PCC) is the international standard approach in dementia care [2,8] and places the person in the center of their own care [9]. This approach posits that people with dementia are first and foremost to be seen as a person with their own history, thoughts, and individual needs [8]. BPSD have multiple causes and can be attributed to individual and environmental factors [10]. Therefore, causes of BPSD are analyzed within a biopsychosocial approach

to emphasize all possible dimensions. It has been well established by various intervention studies that only interventions with a multidisciplinary integrated approach diminish BPSD [11–14]. Various disciplines (such as medicine, psychology, and nursing staff) must work together to develop the personalized integrated treatment in order to address both the underlying causes of BPSD and the residents' individual needs [10,15]. However, multidisciplinary collaboration appears to be difficult to achieve in nursing homes, leading to a smaller decrease in BPSD than expected as a result of low treatment fidelity [12–14].

Early recognition of BPSD is important in order to ensure that patients' personalized integrated treatment begins as soon as possible [16]. However, most residents with dementia are incapable of asking for help due to their cognitive impairment. Physicians and psychologists do not see residents on a daily basis in a nursing home. However, the nursing staff have the opportunity to observe residents' behavior consistently [17]. During care activities, nursing staff make daily observations on the basis of naturally occurring interactions between the residents and staff [18]. Persoon et al. [19] suggest that nurses' daily observations could contribute to the diagnostic process of memory function by identifying residents in need of further neuropsychological assessment. Daily observations may also support early diagnosis of BPSD and clarification of the causes and residents' needs. However, little focus has been placed on studies that focus on the behavioral observations of residents with dementia made by nursing staff. Qualitative research has mainly explored nursing staffs' targeting, ability to cope with, and perspectives on BPSD of residents with dementia [6,7,16,20–22]. The growing number of people with dementia [23], the high prevalence of BPSD [4,24], and the unique position of nursing staff that allows them to continuously observe residents' behavior highlight the need to explore nursing staff's experiences of observing BPSD. To support early identification of BPSD, as well as policy and management practices, this study aims to explore nursing staff's experiences observing BPSD of residents with dementia in a nursing home.

## 2. Materials and Methods

### 2.1. Study Design

To gather in-depth information, a generic qualitative research design was chosen. Semi-structured individual interviews were conducted to allow the participants to express themselves freely and the researcher to seek clarification, ensuring that all of the necessary information was obtained.

### 2.2. Procedure and Participants

The research population consisted of nursing staff caring for people with dementia living in a nursing home. To conduct the necessary behavioral observations, nursing staff had to care for people with dementia on a regular basis. Therefore, inclusion criteria for the purposive sample included working as nursing staff in direct contact with people with dementia in a nursing home, working 16 hours or more per week, having at least one year of experience in caring for people with dementia, and speaking Dutch fluently. The study was conducted in one nursing home with two wards in a homelike environment. The nursing home is part of a care organization with five nursing homes. In this nursing home, there are twenty residents per ward, the majority of which are in the middle stage of dementia, while a few are in the late stage. The nursing home was chosen as it was representative of the majority of nursing homes in the Netherlands, including its use of a PCC approach and the possibility of consultancy of a multidisciplinary team, both adhering to standard procedures [8]. The composition of the nursing staff team corresponds to the common practice in the Netherlands. In Dutch nursing homes, the team consists mainly of certified nurse assistants (CNAs). Only 5% of nursing staff members are registered nurses (RNs) with four years of vocational training obtained via a bachelor's degree. The educational background of nursing staff differs worldwide [25]. In the Netherlands, CNAs receive three years of vocational training and nurse assistants (NAs) receive two years of training. Although RNs, CNAs, and NAs have substantively different tasks

and responsibilities, observing behavior is within the remit of all three [26]. In order to achieve sample variance, nursing staff members were eligible to participate in the study regardless of the participant's level of education in nursing. Participation was voluntary. The study was conducted according to the principles of the Declaration of Helsinki [27] and the guidelines for good clinical practice [28]. Handling and storage of data complied with the Dutch Personal Data Protection Act (www.goverment.nl/privacy). Participants were not subject to treatment, and no code of behavior was dictated; thus, this constituted a non-WMO study, and the local regulations of the institute were followed.

### 2.3. Data Collection and Analyses

An interview guide was used to collect data in semi-structured interviews. The reliability of the interview guide was established during two pre-test interviews (Table 1). This resulted in the reformulation of one question to improve clarity. In the Dutch guideline, BPSD is described as "problem behavior" [8]; therefore, this term was used in the interviews. During the interviews, it became clear that some nursing staff members were more familiar with the term "challenging behavior", and therefore both the terms were used in the interviews.

**Table 1.** Interview guide.

| |
|---|
| *Introduction interview, general questions to enable participant to relax.* |
| • How long have you been working in the area of dementia care? <br> • What nursing qualifications do you have? <br> • Do you have specific dementia qualifications? If yes . . . . what are they? |
| *Main interview, focus on the nursing staff's experiences with observing problem behavior of residents with dementia in a nursing home.* |
| • What behaviors do you consider as problem behavior/challenging behavior? <br> • Can you describe in your own words your definition of problem behavior/ challenging behavior? <br> • Can you tell me how you observe problem behavior/challenging behavior of residents with dementia? <br> • What does observing problem behavior/challenging behavior of residents with dementia mean for you? <br> • What challenges in observing behaviors of people with dementia do you experience? <br> • What is supporting you in observing behaviors of residents with dementia? <br> • How do you consider your ability to observe problem behavior/challenging behavior of residents with dementia? |
| *End interview.* |
| • Do you feel we covered all relevant areas? <br> • Is there anything you would like to add? |

Interviews were conducted in an office within the nursing home to ensure privacy. The data collection was continued until data saturation was achieved, a step that was discussed within the research team. Interviews were conducted between February and April 2019 and were audio recorded. The analysis followed the six phases of thematic analysis, according to Braun and Clarke [29]: (1) familiarizing ourselves with the data through transcription, reading, and re-reading of the data; (2) initial codes were generated of the entire data at a latent level; (3) for collating codes into potential themes, a visual representation was used to support discussion in the research group; (4) an initial thematic map was generated through reviewing all the codes of the potential themes and re-reading and summarizing the whole dataset to examine if they formed a pattern; (5) the final thematic map was developed through a refinement of the names and definitions of the themes in collaboration with the involved researchers; (6) thereafter, the report was formulated by the primary researcher and reviewed by the other researchers. Atlas-ti version 8 was used (www.atlasti.com) to

support the inductive thematic analyses of the data. Data were processed anonymously and stored for 10 years at a secured research site by Rotterdam University of Applied Sciences.

## 3. Results

A rich description of the entire dataset was provided to gain an overall description, which was appropriate for the under-researched subject of the study [20]. The data were analyzed, within a constructionist framework, at a latent level using inductive thematic analysis to emphasize the sociocultural context.

Data saturation was achieved after interviewing 12 nursing staff members, whose details are provided in Table 2.

**Table 2.** Background characteristics of the participants.

|     | Gender | Age (Years) | Primary Qualification | Years Qualified | Years Working in Dementia Care | Dementia-Specific Training |
| --- | --- | --- | --- | --- | --- | --- |
| P1 | Female | 53 | CAN [1] | 33 | 30 | Agogic psychogeriatrics |
| P2 | Female | 61 | CAN [1] | 23 | 20 | |
| P3 | Female | 39 | CAN [1] | 3 | 2 | |
| P4 | Male | 38 | RN [2] | 14 | 13 | |
| P5 | Female | 50 | CAN [1] | 29 | 29 | Agogic psychogeriatrics |
| P6 | Female | 47 | NA [3] | 15 | 15 | |
| P7 | Female | 44 | CAN [1] | 25 | 25 | |
| P8 | Male | 28 | CAN [1] | 3 | 3 | |
| P9 | Female | 25 | CAN [1] | 2 | 1 | |
| P10 | Female | 25 | CAN [1] | 4 | 4 | Agogic psychogeriatrics |
| P11 | Female | 54 | NA [3] | 12 | 12 | |
| P12 | Female | 33 | CAN [1] | 14 | 14 | |

[1] Certified nurse assistant, [2] registered nurse, [3] nurse aide.

The mean duration of the interviews was 56 minutes (range 52–64). Two men and ten women participated in the study. The duration of working experience varied from 1 to 30 years (mean 14 years). Two eligible participants declined to take part in the study, considering the interview to be too demanding. A member check was performed in two different ways. At the end of each interview, the researcher summarized the participants' answers to ensure they were understood correctly. The researcher also presented the thematic map and explained the definitions of the themes to the participants in a team meeting to give participants the opportunity to challenge the results. All the participants recognized the themes.

From the inductive thematic analysis of data, four main themes were identified, containing a variety of sub-themes (Table 3). Both the main themes and sub-themes are explained below and substantiated by quotations from the interviews.

**Table 3.** Main themes and sub-themes.

| Main Themes | Sub-Themes |
|---|---|
| Group harmony | • Viewing the residents as a group<br>• Social interactions between residents |
| Intuitive approach<br>Reactive intervention<br>Sharing information | • Unconscious observing<br>• Not using a method<br>• Detecting triggers<br>• Trial and error determining interventions<br>• Reflection within the nursing staff team<br>• Barriers consulting with a multidisciplinary team |

*3.1. "Group Harmony" Theme*

This theme consists of two sub-themes that describe the way in which nursing staff observed residents from a group perspective to maintain harmony in the group.

3.1.1. Viewing Residents as a Group

Participants' main objective was to create a safe and secure environment for residents with different backgrounds and needs, involuntarily forming a society. Therefore, they considered maintaining group harmony to be their main responsibility in order to ensure "a nice day" for every resident.

*"Peace and quiet of the group are important because, I think for the resident it is very nice if there is calmness. That they really feel at ease, that they are fine and feel safe." (P9)*

Therefore, the behavior of residents was frequently observed from a group perspective. The participants observed the behavior of individual residents next to verbal and nonverbal interactions between residents. Watching over the residents was considered to be an important requirement of nursing staff. The majority of participants described problem behavior as behavior that was disruptive to other residents, family, or staff:

*"I think problem behavior is behavior that is a burden for other people." (P12)*

Participants considered physical and verbal aggression to be the most frequently observed of these behaviors. Only one participant described passive behavior as a form of BPSD.

*"Well, they're going to punch and kick. I also find verbal behavior very bad, that sarcasm." (P6)*

3.1.2. Social Interactions between Residents

When a conflict was observed, participants switched the focus of their observations from the group to the resident or residents disturbing the harmony. Participants also noted that individual disturbing behavior was easily adopted by the group. Uncontrollable escalation within the group was perceived as a significant danger that must be prevented. Thus, participants deemed it important to observe residents' physical, verbal, and non-verbal interactions so that they could intervene quickly:

*"There was a lady who interfered with all the residents around her. Then she says yes, but I must go home. Then she has forgotten that she lives here. But then she goes to other residents, I want to go home, do you want to go home too? ... That goes on and on, and then it goes on like a wildfire over the department. ... You have to stop that, otherwise everyone wants to go home." (P8)*

### 3.2. "Intuitive Approach" Theme

The intuitive approach describes how observations of residents' behavior were obtained in two sub-themes: unconsciously and without using a method.

### 3.2.1. Unconscious Observing

The participants were unaware of the way in which they observed behavior. They explained that observation was not an activity on its own; rather, it occurred during caregiving. As a result, participants were unable to describe the way in which they made observations. Some participants explained their ability to observe as a natural talent or gift:

> *"That actually goes unnoticed. Actually, you scan the whole day to see what happens. You are busy here and you can immediately see what happens in another corner. I think I have that as a gift, a bit. Yes, yes." (P1)*

### 3.2.2. Not Using a Method

Participants did not use a particular method to observe residents' behavior; rather, they considered the observation of BPSD behavior to be a skill learned through practical experience:

> *"No, it's really just watching how someone responds. I just talked it over last time with my colleague; I find it so special that someone no longer knows who his children are... But, I take no time to look for the theory behind it." (P9)*

All participants unanimously agreed that no focus was placed on the theory of behavioral observation during their nursing education. Some participants doubted that knowledge could support the observation of behavior.

### 3.3. "Reactive Intervention" Theme

Reactive intervention refers to how nursing staff detected triggers and how they determined interventions.

### 3.3.1. Detecting Triggers

Participants observed the triggers of problem behavior, defining a trigger as a cause of a behavioral change. Such triggers included noise, crowds, a specific person, or physical discomfort. Participants found it important to know the residents in order to distinguish normal and abnormal behavior. The participants searched for triggers by repeatedly observing the residents and identifying the relationship between the change in behavior and the provoking factor. The determination of a trigger was achieved through its instant removal, without exploring the origin of the resident's problem behavior. Participants considered this instant intervention to be necessary:

> *"Sometimes you see it after family visits. That sad lady who stays with us, when her husband comes over, then she wants to go home . . . Then we know, if he doesn't leave soon, we will enjoy it for a few hours. She is restless, angry, cries a lot, throws her bag on the floor . . . while if you distract her for a moment and he goes away quickly, that's a lot less." (P5)*

### 3.3.2. Trial and Error Determining Interventions

Participants often mentioned distraction, moving a resident to a quiet place, and giving personal attention as suitable interventions to eliminate a trigger. The most effective intervention for an individual resident was discovered by trial and error after observing the effect of the intervention:

*"It's just trial and error... with the magic table or watching television. Very often it is trying to see if something works . . . watch how they react to it." (P5)*

Participants observed changes in the behavior of residents after the intervention. However, there was no deliberation or analysis before and after the intervention to determine the appropriate measures for individual residents on the basis of the cause(s) of the behavior.

*"That lady who is so angry, can swear so much. Sometimes we know what makes her angry, for example if she is incontinent of stools..... But I find it difficult, how do you look deeper, yes I can't describe that well. That's a feeling too." (P7)*

*"When I 'am sitting in the living room with a resident who gets very angry with a certain person, I try to get them out of there. Then at least they don't have that stimulus that can irritate them." (P9)*

### 3.4. "Sharing Information" Theme

Nursing staff shared information about observed behavior within and outside the nursing team, which is described in the two subthemes below.

### 3.4.1. Reflection within the Nursing Staff Team

The participants considered an experienced nursing team to be responsible for and able to solve BPSD behavior. Participants were aware that their approach influenced the behavior of residents, and therefore they felt responsible:

*"One resident shouts a little more and the other demands a little more and the other is a little more aggressive... because some residents you can't change... And if it gets [to be] a burden for them, I think we should act." (P4)*

*"I discuss a client in the multidisciplinary team when it is above my head. You just want to get it done yourself... It's kind of a failure that you can't get it done. That you did, you just learned so many things and so much experience and then you don't know it anymore. I think it's failure, kind of not wanting to fail." (P12)*

Nursing staff reflected with each other on whether their approach in fact diminished or triggered BPSD. As the behavior occurred, they discussed it amongst themselves, but this process lacked analysis or evaluation. Participants explored the behavior from different perspectives and exchanged experiences to find agreement in ways of care:

*"Well, for example, the screaming of a resident, which can chill you to the bone. Where everything has already been tried, approach, medication, distraction, really everything you can think of . . . .. Additionally, those are things that are often discussed within the team. How do you do that or what works for you." (P9)*

3.4.2. Barriers Consulting the Multidisciplinary Team

Although consultation of the multidisciplinary team was possible for nursing staff, they experienced barriers to achieving this in good time. Participants mentioned that they only consulted the multidisciplinary team when it was beyond their capabilities to resolve the BPDS behavior of the resident. Participants gave different reasons why they did not feel facilitated to share their observations with other disciplines. In their opinion, it was difficult to share and explain observed behavior, as understanding could only be achieved through experience.

> *"It is sometimes difficult to properly explain to a doctor and sometimes a psychologist what exactly the behavior is. We once had a man who went crazy in the evening, a real problem. The psychologist has already gone home by then and then it is difficult to make clear what it is really doing on the ward." (P5)*

Many participants felt that they were not taken seriously by other disciplines.

> *"Yes when you say [in the multidisciplinary team] that the same solution is not being invented for the umpteenth time. Some things we have already tried a hundred times. We see the people every day, it does not work. If you then say that we have already tried and it is then said [by the psychologist] that I want you to try it anyway, then it is as if they do not believe us." (P9)*

The majority of participants were not confident that the multidisciplinary team could help to reduce the residents' BPSD behavior. However, some participants mentioned that the multidisciplinary team provided useful recommendations:

> *"I think the psychologist who has no experiences with daily care... they just see pieces of challenging behavior. Maybe they lack the experience to fully understand the resident, or something. They also rarely come up with ideas which help us." (P6)*

## 4. Discussion

In this study, we identified four main themes. Nursing staff observed the BPSD of residents from a group perspective, focusing on those who were disturbing the harmony with the aim of intervening in a timely manner; this related to the "group harmony" theme. Observations were made using an intuitive approach, unconsciously and without a method; this followed the "intuitive approach". A reactive intervention was used, which comprised immediate removal of the observed triggers without investigation of the under-lying cause(s) of BPSD; this was referred to as the "reactive intervention" theme. Nursing staff shared information within the team itself and felt responsible for solving the BPSD. Therefore, they only consulted the other members of the multidisciplinary team when the situation became unmanageable; this complied with the "sharing information" theme.

The theme "group harmony" describes how nursing staff mainly observed residents from a group perspective, focusing on physical, verbal, and non-verbal interactions between residents. Surprisingly, this novel theme was identified from the data and has not previously been mentioned in the literature, to our knowledge. Nursing staff predominantly observe social interactions between residents to sustain a safe environment for residents living together in a nursing home. Previous research has identified the relevance of social contact between residents. Residents are confronted with other residents' BPSD; this is often perceived as an alienating or even a frightening experience for residents [30]. Nurses can stimulate positive interaction between residents and function as social-action agents [31,32]. Thus, nursing staff's efforts to maintain harmony in the group may contribute to a positive relationship between residents, improving the perceived well-being of the residents. At the same time, nursing staff's focus on group harmony meant that they mainly observed aggression and agitation, because these behaviors disrupted the harmony of the group. This finding may explain why other studies found "quiet" BPSD such as apathy and depression are rarely identified by nursing staff [6,33].

The "intuitive approach" theme describes how observations of residents' behavior were obtained, unconsciously and without using a method. The nursing staff were unable to describe how they made observations because observing was not considered to be an activity in its own right. Some nursing staff members stated that observing behavior was a talent, and they all found it to be a skill learned through practice. Intuition is an important type of nursing knowledge displayed by experienced care professionals [34]. Intuition helped nursing staff to observe a part of BPSD. Caring is defined by Leninger as "actions and activities directed towards assisting, supporting, or enabling another individual or group with evident or anticipated needs to ameliorate or improve a human condition or lifeway, or to face death" [35]. This definition is in line with the PCC approach to people with dementia, focusing on needs. Next to intuitive observations are daily observations of behavior made by nursing staff. These are especially useful when it comes to investigating the unmet needs of residents. McCabe [36] and Steele [15] proved that a structured way of observing helps nursing staff to identify the unmet needs of people with dementia and understand BPSD. There are several methods nursing staff can use to structure their daily observations. We argue for a method that is in line with the biopsychosocial model to address the multifactorial causes of the BPSD and to sustain interdisciplinary collaboration. The Dutch guideline of problem behavior is based on the biopsychological model [8]. A biopsychosocial approach aims to formulate a person's condition and subsequent care in terms of their psychological (emotional and behavioral) and socioenvironmental (cultural) factors, as well as pathology [32]. Examples of methods used to structure daily observations include the Dynamic System Analyses method (DSA) method [37], the VIPS framework (valuing people, individual lives, personal perspectives, social environment) [38], and the Newcastle Model [15].

The third theme, "reactive intervention", revealed how nursing staff detected triggers and the way they determined interventions. The observed triggers that provoked the disruptive BPSD were immediately removed using practical interventions based on trial and error. This prohibited nursing staff from investigating the underlying reasons for the residents' BPSD. Zeller [16] also found that nursing staff observe triggers of BPSD but do not apply a systematic approach. The practical interventions nursing staff used to remove the observed triggers, such as distraction, personal attention, and bringing the resident to a quieter place, reflect an attempt to fulfil an individual need expressed by the resident. Nursing staff believe they are using a person-centered approach by deploying reactive interventions. However, by removing the trigger, nursing staff did not explore or fulfil the needs of the resident. While these interventions restored the harmony in the group and diminished the BPSD for a short period, they did not reveal the cause(s) or individual needs of the resident in question. However, to diminish BPSD in the longer term, personalized integrated treatment is required; thus, intervention should be based on the underlying cause[s] and the residents' needs [10–12,14]. Thus, the cause(s) of the BPSD and the individual residents' needs must be addressed rather than the behavior itself.

The last main theme, "sharing information", revealed nursing staff shared information about observed behavior within the nursing team itself but experienced barriers when consulting the other members of the multidisciplinary team. Nursing staff only shared observed BPSD of the residents within the multidisciplinary team when it was beyond their capabilities to restore the harmony of the group. They felt responsible for resolving the BPSD by themselves. In our study, the majority of nursing staff felt that they were not taken seriously by other disciplines. The nursing staff had differing opinions about the usefulness of the recommendations provided by the multidisciplinary team. Several studies have shown that to reduce BPSD, a methodical multidisciplinary integrated approach targeting the cause(s) of the individual resident is needed [10,11,13,14]. In order to achieve a personal integrated treatment, nursing staff must share their observations of residents' BPSD within the multidisciplinary team according to their own expertise. As stated previously, methodological observations during caregiving of nursing staff are especially suitable for identifying the needs of the individual resident with dementia. This perception

of nursing staff aids discussion in the multidisciplinary team following a PCC approach, focusing on the person and not just the disease.

Nursing staff indicated they found it difficult to share and explain observed behavior with other disciplines, a statement supported by the literature [39]. Additionally, physicians often find nurses' observations of BPSD to be insufficient [40]. It is a matter of concern that physicians simultaneously rely on the observations of nursing staff and wait for the nursing staff to consult them [40,41], especially because this study revealed that nursing staff only consult a multidisciplinary team when the BPSD of a resident become unmanageable. Physicians and psychologists should therefore also act proactively and follow the clinical diagnostic phases for identification, description, and clarification of the problem, as well as accomplish additional research as described in the guideline [5]. Additionally, an interprofessional collaboration can support timely communication between all disciplines within the multidisciplinary team. Interprofessional collaboration is defined by Tsaki-tzidis et al. [41] as a model of collaboration between different healthcare providers. It requires awareness of different roles in the team, a common vision and goals, a purposeful approach, and a sense of shared responsibility [39]. A large amount of collaboration is perceived when providing standard care in nursing homes, but without a clear role description of the different disciplines [41].

The results of how nursing staff observe BPSD provide an insight into the reasons for the low treatment fidelity found in many intervention studies [12–14]. There is a growing body of literature that recognizes that the use of a methodical approach to ob-serve and analyze causes of the BPSD of people with dementia requires knowledge and skills [7,14,42,43]. Therefore, nursing staff need to be educated to methodologically ob-serve the behavior of residents with dementia. Additionally, open, timely communication between all disciplines is essential for interprofessional collaboration within a multidis-ciplinary team. The way nursing staff members perform their observations is significant because optimal information is a requisite to improve treatment fidelity and achieve an effective personalized integrated treatment.

This study has some limitations. It was conducted in a single nursing home, and although this nursing home was chosen because it was representative of the majority, this may affect the transferability of the results. Further research into the experiences of nursing staff observing BPSD in other nursing homes with different care settings, such as small living facilities and large psychogeriatric wards, may be warranted. Observations of behavior of residents may differ between care settings; however, the prevalence of BPSD is equal [43]. Further, data were collected only through individual interviews, making it difficult to determine whether participants' claims reflect their actions when observing residents. Still, a strength of this study is that participants expressed themselves freely, which provided a new understanding of the ways in which nursing staff observe residents from a group perspective.

## 5. Conclusions

In general, the four themes identified in this paper are important barriers to the implementation of personalized integrated treatment for BPSD of residents with dementia living in a nursing home. Nursing staff mainly observe behavior from a group perspective. Since nursing staff generally focus their observations on the disturbance of group harmony, less focus is placed on observing the needs of the individual resident with dementia. In addition, nurses tend to keep their observations to themselves, while physicians wait for the nursing staff to consult them. Improvements in interprofessional collaboration are needed to encourage nursing staff to share their observations of behavior in a timely manner. The cause(s) of the BPSD are not explored by nursing staff as they generally do not observe behavior methodologically and use reactive intervention based on immediate removal of the observed triggers. Training nursing staff to use a specific method when making daily observations can empower nursing staff to share their observations within the multidisciplinary team on the basis of their own expertise. The perception of nursing

staff helps with discussion in the multidisciplinary team following a PCC approach and centers the resident's needs. This is important because optimal information is a requisite to improve treatment fidelity and achieve an effective personalized integrated treatment.

## 6. Recommendations

Nursing staff must be taught to use a method, in line with the biopsychosocial model, in order to structure their daily observations of behavior and assess the possible causes of BPSD. A practical and feasible way to implement working with a method that's suits nursing staff is training on the job. In order to empower nursing staff to effectively share the observed behavior from their professional perspective within the multidisciplinary team, it is necessary to improve interprofessional collaboration in nursing homes, beginning with a clear role description of the different disciplines.

**Author Contributions:** Conceptualization, E.G., I.P. and T.B.; data collection, E.G.; analysis, E.G., I.P. and T.B.; writing first draft, E.G.; rewriting and editing, E.G., I.P. and T.B. All authors have read and agreed to the published version of the manuscript.

**Funding:** This research received no external funding.

**Institutional Review Board Statement:** Not applicable.

**Informed Consent Statement:** Informed consent was obtained from all subjects involved in the study.

**Data Availability Statement:** Data used and analyzed in this study will be promptly available for the publisher upon request.

**Acknowledgments:** We are grateful for the nurses who took part in the study.

**Conflicts of Interest:** The authors declare no conflict of interest.

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
