# Peer review of "Nursing Staff’s Observations of BPSD Amongst Older Adults with Dementia Living in a Nursing Home: A Qualitative Study"

_nursrep, doi:10.3390/nursrep13010018_

Round 1

Reviewer 1 Report

The article focuses on Nursing Staff’s Observations of Challenging Behavior Amongst Older Adults with Dementia living in a Nursing-home

The article is well structured and there is alignment between the various sections; well-founded and well-referenced; 

I congratulate the authors. 

Some observations/suggestions for improvement follow below, please consider:

Keywords – Suggestion: removed the number near each keyword;

challenging behavior – it is not MeSH term;

In the Abstract, Introduction, and throughout the various sections of the article, the authors use the term “Challenging behavior".

Indeed, the authors might have the motivation to call “Challenging behavior”, but I have difficulty in understanding. Because there is a name for this challenging behavior - psycho-behavioural symptoms or neuropsychiatric symptoms. Why do authors give another name?

Title: Suggestion - Remove "matters"

Nursing Staff’s Observations of Challenging Behavior Amongst Older Adults with Dementia living in a Nursing-home: A Qualitative Study

Introduction -The term "challenging behavior" is not defined. It is important to clarify this concept ...

I might even agree with the use of the term "challenging behavior" in the methodology and results section; given that the interviews must be open and the "non-use of the terms psycho-behavioral symptoms or neuropsychiatric symptoms" can "open" the range of answers. But in fact it didn't even happen, because when analyzing the results, these psycho-behavioral symptoms are listed (agitation... apathy...)

In conclusion, I can agree with the use of "Challenging behavior" in the title and results, but in the introduction, discussion  and conclusion it seems reductive to use this concept.

Suggestion:  the introduction a link between the concept "Challenging behavior" and psychobehavioral symptoms should be made.

Especially because the authors mention very well the need for work between different members of the team and that we have to support teamwork. For this, the use of a language understandable by all is essential, creating new concepts will not help teamwork.

Therefore, I think that by introducing a new concept, we are also not being professionals or knowing how to communicate with other health professionals from other disciplines.

This study is essential and should be read by health professionals other than nurses and also by nurses. Will nurses know what challenging behavior is in the area of dementia?

I notice that they also use the bibliography of  papers on psycho-behavioral symptoms to substantiate these "challenging behavior", eg: reference 8, ...

Method

To encourage the replication of this study in other Nertherland units and in other realities, the interview guide, that is the main questions, could be incorporated into the paper, more specifically in the subsection - Data collection and analyzes

The characterization of nursing home could enrich the paper, if it is still possible. How many older adults did this nursung home have? and of these, how many would have cognitive deterioration... and to what degree (these are important indicators that can influence the findings found)

Results

The beginning of the results section could be structured in a more reader-friendly way. Note that they refer to table 2 - Line 107 to 108 "Data saturation was achieved after interviewing 12 nursing staff members, whose details are provided in Table 2."; but they do not refer to table 1 in the body of the paper. 

It is advised to be carried out.

Suggestion: Table 2 remove “ Tables may have a footer”

Standardize the use of capital letters, for example: line 264 - The theme “group harmony” describes nursing staff mainly observed residents ; line 284 - ‘Intuitive approach’ ; as well use of the  ' or ''

Line 318 . of ?

Use [] for citations. e.g: [2]

Conclusion

What are the practical implications of your findings (despite making a small mention, there is room for improvement); and what are the implications for education/teaching and research? 

Congratulation

Reviewer 2 Report

Thank you for the opportunity to review this manuscript. The article has the potential to add to nursing knowledge, but there are weaknesses that should be addressed to strengthen this paper.

Abstract: The abstract was informative but contained grammatical errors. For example, on page 1, line 16, the sentence 'a generic qualitative design as chosen' is grammatically wrong. The conclusion reached in the abstract and the article is also problematic, but this will be expanded on later.

Introduction: The introduction was well crafted and enjoyable to read. However, there were many typos and grammatical errors, such as on page 1 (line 32 - depend not dependent, lines 42-43 should list the discipline as medicine, psychology). Page 2, line 67, 'the aim of, not 'o'. Please do thorough language editing.

Materials and methods:

Study design: Lines 74 to 77 on page 2 should be moved to the analysis section.

Procedure and participants: Line 81, page 2, change 'base' to basis. Kindly clarify whether ethics approval was received for the study.

Data collection and analyses: Pre-test is a more appropriate terminology to pilot.

Result: Although the participants were described as nursing staff, only one participant was a registered nurse. Considering that CNA, RN and NA have different scopes of practice, it would have been interesting to disaggregate the findings. For example, a nurse aide, might observe, but not be able to intervene therapeutically due to the scope of practice and may also not report to RNs who have a broader scope of practice. It would be important to provide more context on the different scopes of these specialisations and give a rationale for why RNs were under-represented in the study. 

3.2.2 (not using a method): Line 189 to 192 is unclear and should either be deleted or moved to the first theme as it is about group observation. Overlooking itself is not clear. Do you mean a group of residents were being overlooked or that staff watched over the group?

Other comments on the result section: There were many references to harmony, even in theme three, which was about reactions to challenging behaviours. For example, under the subtheme 'detecting triggers', the quotation used to support this subtheme was of an individual who was triggered by her husband's visit, and the intervention was individualized, not group-based. How, then, does this fit in with group harmony? Also, the statement at the beginning of the quote, which was the question used to elicit the quote, 'can you give another example of a trigger', should be removed.

3.4.2: This subtheme should be supported with appropriate quotes from the data. As previously highlighted, the paper was positioned in a way that suggests that all the observations occur in a group, but some quotes suggest otherwise. It is also unlikely that observations take place only in group settings. This perception needs to be revisited. The quote supplied under 3.4.2 is not a barrier to consulting. Rather, it suggests that MDT is inexperienced or does not help. But the access was there.

Discussion: The discussion and the conclusion reached are problematic in many ways and will be explained below:

1. On page 7, line 280, it was stated that staff overlooked 'quiet behaviours like apathy and depression. This conclusion is beyond the aim of this study which was to 'explore nursing staff's experience with observing challenging behaviour of residents with dementia in a nursing home'. The diagnosis of depression, for instance, is not reached through observation or observation alone.

2. On page 7, lines 285 - 286, it was stated that nursing staff could not describe how they made observations, and it recommended a model or structured way of observing. This perception does not recognise the ways of nursing (Carper, 1978; Chinn & Kramer, 2008). In nursing, there are four valid ways of knowing (empirical knowledge, aesthetics, personal knowledge and ethical knowledge). Intuition is a valid form of knowledge and knowing, and it comes with experience. A nursing staff does not need to be able to explain how they knew a patient is having challenging behaviour. As such, the conclusion reached in this article does not reflect the breadth of nursing knowledge and as such, the conclusion is oriented within a medical model that sees the world from a cause-and-effect perspective. 

3. What is the rationale for the bio-psycho-social model? The data does not support this, and there should be a well-reasoned justification for this.

4. The data did not support other assumptions on page 7 (lines 319 and 325) that suggest nursing staff differed in opinion and that they found it difficult to share and explain the observed behaviour. It could be your interpretation of the data, but this was not clear from the data.

5. Some sentences are unclear in the discussion. For example, see page 8 (lines 329 to 331, lines 341 - 345). Please revise for grammar and clarity.

6. Please revise the whole transcript for omissions, grammar and clarity

Round 2

Reviewer 2 Report

Thank you for addressing the suggestions. I am satisfied with the changes made. In the manuscript, the acronym RA was used for registered nurses. I am not quite sure how this was arrived at. Also, as some of the interview questions asked participants their definitions of challenging behaviour, it would have been interesting to see a theme about that. For example, whether variations exist among their conceptualisation of the term and how it compares with other established definitions of challenging behaviour within dementia. This would also have helped the study establish the definition of challenging behaviour. 
